# Understanding Retrieval Augmentation for Long-Form Question Answering

**Hung-Ting Chen, Fangyuan Xu**\*, **Shane Arora**\*, **Eunsol Choi**
Department of Computer Science
University of Texas at Austin
{hungtingchen,fangyuan,shane.arora,eunsol}@utexas.edu

## Abstract

How retrieved documents are used in language models (LMs) for long-form generation task is understudied. We present two controlled studies on retrieval-augmented LM for long-form question answering (LFQA): one fixing the LM and varying evidence documents and the other fixing evidence documents and varying the LMs. We study various attributes of generated answers (e.g., fluency, length, variance), with an emphasis on the attribution of generated answers to in-context evidence documents. We collect a dataset (SALAD) containing human annotations of sentence-level answer attribution in LFQA and evaluate existing methods for automatically judging attribution. We find that while LMs can leverage relevant in-context documents, the generated answer is only partially attributable towards the documents, especially for LMs trained without retrieval augmentation. Together, our analysis reveals how retrieval augmentation impacts long knowledge-rich text generation and provide directions for future work.

## 1 Introduction

Recent works (Nakano et al., 2021; Malaviya et al., 2023; Gao et al., 2023b) proposed retrieval augmentation as a powerful tool to provide up-to-date, relevant information to LMs for long-form answer generation. Yet, retrieval augmentation does not always affect LMs the way we anticipate. Liu et al. (2023a) discovered that information placed in the middle of contexts is not used by LMs. Parametric knowledge continues to affect generation even when only relevant documents are provided in-context for factoid QA task (Longpre et al., 2021; Chen et al., 2022). These findings, however, are based on analyses on factoid QA with short answer spans, which is easier to evaluate. Our understanding of how retrieval augmentation impacts long-form generation in LMs is limited.

We study how retrieval impacts answer generation for LFQA, a complex long-text generation task. We study two settings (illustrated in Figure 1): (1) fixing the LM and varying the degree of relevance of evidence documents and (2) fixing evidence documents and varying the LMs. As evaluating the quality of long-form answers is notoriously difficult (Krishna et al., 2021), we start our analysis by measuring surface features (e.g. length, perplexity) that correlate with specific answer qualities such as coherence (Xu et al., 2023).

Our analysis reveals that retrieval augmentation changes LM's generation substantially. Some effects, e.g., change in the length of answers, are pronounced even when provided documents are irrelevant. Relevant in-context evidence documents lead to more substantial changes, leading LMs to generate more unexpected sentences (measured by higher perplexity), while irrelevant documents do not have the same effects. Surprisingly, the impact of retrieval augmentation, even with the same set of evidence documents, can result in opposite effects for different base LMs (e.g. change in answer lengths and lexical diversity).

One desirable property of retrieval-augmented LFQA system is to have the answer *attributed* to the evidence documents. To evaluate this, we collect SALAD, a dataset with **S**entence-

---

\*Equal contribution

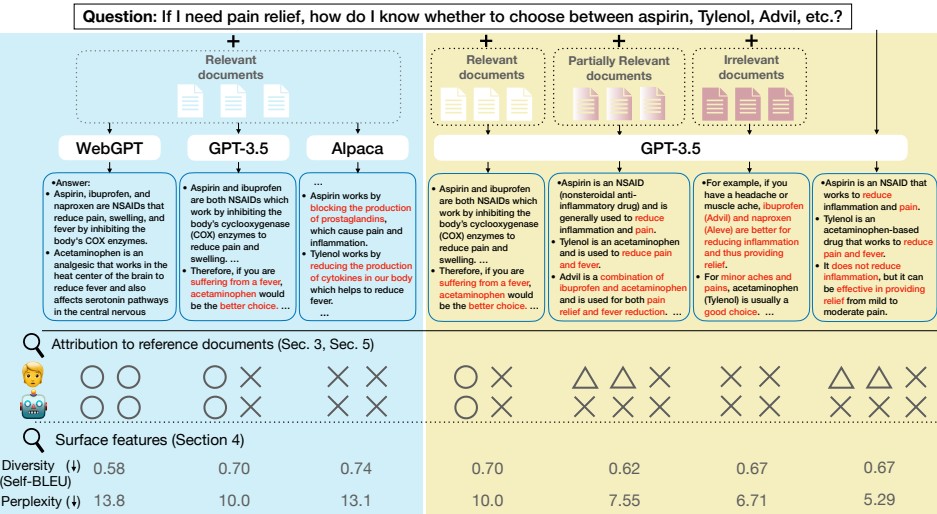

Figure 1: We study (A) how differing LMs use the same in-context evidence documents to generate answer and (B) how in-context documents of various degree of relevance affect the answer generation. We analyze generated answers on surface patterns (self-bleu, perplexity, etc) and their attribution to evidence documents. Attribution judgements are made per sentence, either by annotators (Section 5) or automatically from NLI model (Section 7). O's, Δ's and X's denote supported, partially supported and unsupported sentences respectively. Colored texts are generated texts not supported by in-context evidence documents.

level **A**ttribution of **L**ong-form **A**nswers to evidence **D**ocuments, 12k human attribution judgements over 4k answer sentences. We provide an in-depth analysis of attribution with SALAD, which can serve as a benchmark for evaluating automatic attribution. We observe NLI models that performed well in detecting attribution in factoid QA (Bohnet et al., 2022) perform competitively in LFQA setting as well. They significantly outperform chance baseline yet fall behind human agreement by 15% in accuracy. Our study reveals that attribution quality varies greatly across base LMs, even when they are provided with the same set of documents.

Our analysis reveals new insights on attribution patterns for long-form generation. The last generated sentence is substantially less attributable than earlier ones, and the generated text tends to follow the order of the in-context evidence documents, even when the in-context document contains multiple concatenated documents. Taken together, our study improves the understanding of how LMs use in-context evidence documents for LFQA and suggests concrete directions for future work. Our data and code is available at https://github.com/timchen0618/LFQA-Verification.

## 2 Background and Related Work

**LFQA** LFQA (Fan et al., 2019; Stelmakh et al., 2022) requires models to generate paragraph-length answers to complex, open-ended questions. To address this, WebGPT (Nakano et al., 2021) introduces a web agent that searches the web and integrates relevant information to LMs. We evaluate this model closely in this study.

**Retrieval Augmentation** Retrieval-augmented generation has received attention as a way to provide up-to-date, relevant documents to LMs at inference time (Ram et al., 2023), showing performance gains across multiple tasks (Shi et al., 2023). A line of work investigates *how* LMs incorporate in-context documents (Mallen et al., 2023; Liu et al., 2023a) with their parametric knowledge on simpler tasks such as factoid QA. Wang et al. (2023) studies the impact of retrieval in open-ended text generation with kNN-LM (Khandelwal et al., 2019).

We focus on LFQA, which requires factual, attributable output over long sequences. Prior work (Krishna et al., 2021) also analyzed attribution in LFQA but studied smaller LLMs fine-tuned with in-domain data, whose attribution pattern varies significantly from the models we study.

**Evaluating Attribution**   We focus our analysis on the attribution of long-form answers with respect to the prepended evidence document set. We follow the AIS framework Rashkin et al. (2021), an evaluation framework for whether a system-generated text can be derived by a given knowledge source. Bohnet et al. (2022) and Yue et al. (2023) study automatically evaluating attribution; the former uses off-the-shelf entailment models, while the latter prompts and fine-tune LLMs. Gao et al. (2023b) builds QA models that generate text along with citations and evaluates the citation quality of the generations automatically. Bohnet et al. (2022) presents a controlled study of attribution (e.g., varying evidence documents and how they impact attribution) on factoid QA with Wikipedia as retrieval corpus.

Recent work (Liu et al., 2023b) annotates attribution quality in long-form answers generated from commercial generative search engines. While they provide a comprehensive study on attribution quality with manual annotations, their study on black box models is limited, as they do not have knowledge of how the cited documents were integrated into the LMs. For instance, documents could have been retrieved post hoc (Gao et al., 2023a) or prepended in-context. We instead present a controlled study involving open-source models, and analyze their data in Section 6.

## 3   Study Setting

We conduct a controlled study on how retrieval augmentation impacts long-form answer generation for LMs, observing surface features and attribution while varying evidence document sets and LMs. In this section, we describe our experimental setting.

**Dataset**   We source questions from ELI5 dataset (Fan et al., 2019), which contains questions from the Reddit forum "Explain Like I'm Five". We use the entire test set released by WebGPT (Nakano et al., 2021) (271 questions) for automatic evaluation (Sec. 4 and Sec. 7.2), and randomly sample 100 questions to collect manual attribution annotations (Sec. 5).

**Knowledge Source: Evidence Documents**   For each question, we compile four sets of evidence documents to examine how models use documents of varying degrees of relevance. Each document set $D$ contains 3-4 document snippets, each containing roughly 100 words. The statistics on each set can be found in Appendix A.1. We detail each document set below:

- **Human Demonstration** Annotators from prior study (Nakano et al., 2021) used a commercial search engine (Bing) to gather documents to answer questions. We include these as "gold" documents that are considered relevant for answering questions by humans.
- **WebGPT Retriever** We consider documents retrieved by the WebGPT (175B) (Nakano et al., 2021) model. Their study found using these documents results in high-quality answer generation.
- **Bing Search** We retrieve relevant documents using Bing Search API with the question as the query, and obtain the top 10 pages returned by the API, and retrieve four 100-word segments from aggregate search results. Post-processing details are in Appendix A.2.
- **Random** To simulate a set of irrelevant documents, we randomly sample another question in the test set and take the corresponding documents retrieved by WebGPT.

We evaluate the relevance of the first three sets of documents manually by sampling 20 questions and examining the document sets. We find that WebGPT, human demonstration and Bing documents contain sufficient information to answer the question for 85%, 50% and 45% of the examples, respectively. Details on the manual study are in Appendix B.6.

**Base LMs & Answer Generation**   We consider three LMs: WebGPT(175b) (Nakano et al., 2021), GPT-3.5 (text-davinci-003) (Brown et al., 2020) and Alpaca (Taori et al., 2023). The

| Model (+ evidence) | # Sentences | # Words | RankGen (↑) | Self-BLEU (↓) | Perplexity (↓) |
|---|---|---|---|---|---|
| **WebGPT**(+ WebGPT docs) | $6.7_{-/1.9}$ | $160_{-/33}$ | $11.35_{-/1.98}$ | $0.58_{-/0.07}$ | $13.81_{-/4.86}$ |
| **GPT-3.5** | $9.3_{1.5/2.6}$ | $219_{30/51}$ | $12.77_{0.67/1.87}$ | $0.71_{0.04/0.06}$ | $6.13_{0.02/1.37}$ |
| **+Human docs** | ${\color{red}6.6}_{0.9/1.8}$ | ${\color{red}172}_{18/40}$ | ${\color{red}11.89}_{0.60/1.86}$ | ${\color{red}0.62}_{0.04/0.07}$ | ${\color{blue}10.94}_{0.05/3.94}$ |
| **+WebGPT docs** | ${\color{red}6.8}_{0.9/1.8}$ | ${\color{red}185}_{20/41}$ | ${\color{red}11.97}_{0.60/1.79}$ | ${\color{red}0.62}_{0.04/0.07}$ | ${\color{blue}11.63}_{0.13/4.16}$ |
| +Bing docs | ${\color{red}6.9}_{1.0/1.9}$ | ${\color{red}179}_{19/38}$ | ${\color{red}12.13}_{0.68/1.91}$ | ${\color{red}0.64}_{0.04/0.07}$ | ${\color{blue}9.03}_{0.12/3.24}$ |
| +Random docs | ${\color{red}7.6}_{1.1/2.1}$ | ${\color{red}183}_{19/39}$ | ${\color{red}12.40}_{0.67/2.13}$ | ${\color{red}0.68}_{0.04/0.07}$ | ${\color{blue}6.76}_{0.05/1.86}$ |
| **Alpaca-7b** | $5.0_{1.8/8.1}$ | $113_{33/73}$ | $12.17_{0.72/2.00}$ | $0.51_{0.09/0.15}$ | $11.95_{0.02/7.18}$ |
| +Human docs | ${\color{blue}5.7}_{1.9/3.6}$ | ${\color{blue}138}_{44/79}$ | ${\color{red}11.82}_{0.88/2.32}$ | ${\color{blue}0.55}_{0.09/0.14}$ | ${\color{blue}12.99}_{0.20/5.73}$ |
| **+WebGPT docs** | ${\color{blue}6.2}_{2.3/7.9}$ | ${\color{blue}145}_{45/80}$ | ${\color{red}11.91}_{0.75/2.07}$ | ${\color{blue}0.55}_{0.08/0.14}$ | ${\color{blue}13.27}_{0.13/5.68}$ |
| +Bing docs | ${\color{blue}7.6}_{2.8/5.0}$ | ${\color{blue}187}_{66/107}$ | ${\color{red}12.04}_{0.78/2.05}$ | ${\color{blue}0.59}_{0.08/0.14}$ | ${\color{red}10.81}_{0.13/5.34}$ |
| +Random docs | ${\color{blue}5.2}_{1.6/5.3}$ | ${\color{blue}121}_{32/65}$ | ${\color{blue}12.25}_{0.71/1.99}$ | ${\color{blue}0.53}_{0.08/0.14}$ | ${\color{red}11.92}_{0.23/5.35}$ |
| Human(+ Human docs) | $5.1_{-/2.7}$ | $119_{-/59}$ | $9.29_{-/4.37}$ | $0.49_{-/0.17}$ | $17.63_{-/7.53}$ |

Table 1: Generated answer statistics. We present means and two standard deviations in its subscript: one computed over three answers generated for the same example, one over answers for different examples. Numbers in red and blue indicate decrease and increase from the base model respectively. We **boldface** rows where we collect human annotations for attribution (Section 5).

WebGPT model is trained to interact with a commercial search engine (Bing) and compose long-form answers based on the information gathered from Bing for questions from the ELI5 dataset.[1] We experimented with a range of open-sourced LMs (GPT-J (Wang & Komatsuzaki, 2021) (6B), Flan-T5 (Chung et al., 2022) (11B), Llama (Touvron et al., 2023) (7B, 13B, 30B) and Alpaca 7B (Taori et al., 2023)) and found Alpaca to be the best-performing upon manual examination.[2] The prediction examples for all other LMs can be found in Appendix B.5. We prepend the concatenated evidence document set to the question and provide it as a prompt to LMs with a brief instruction. We sample three answers for each setting to study answer variability. The decoding hyperparameters and prompts can be found in Appendix A.3.

## 4  How In-Context Documents Impact Surface Answer Statistics

**Metrics**   Unlike evaluating short, mostly entity-based answers (Rajpurkar et al., 2016; Fisch et al., 2019), evaluating the *overall* quality of long-form answers (Krishna et al., 2021; Xu et al., 2023) is notoriously difficult for both humans and models. In this section, we look at metrics that have been shown to correlate with specific aspects (e.g., fluency, coherence) (Xu et al., 2023) of answers, to quantify differences between answers.

- **Length**: We report the number of sentences and number of words in the answer paragraph. The length is shown as a significant confounding factor in human evaluation for various tasks, with humans often preferring longer answers (Sun et al., 2019; Liu et al., 2022; Xu et al., 2023).
- **Self-BLEU (Zhu et al., 2018)** measures the lexical diversity of generated text. An answer is less diverse and contains more repetition if it has a higher Self-BLEU score. Prior work Xu et al. (2023) also found that a lower Self-BLEU correlates to better coherence.
- **RankGen (Krishna et al., 2022)** is a T5-XXL encoder trained with large-scale contrastive learning, ranking generation given a prefix. A higher RankGen score signifies a more likely continuation of the prefix. We measure RankGen score with question as the prefix.
- **Perplexity**: We report the perplexity of the answer measured with GPT-2-XL (Radford et al., 2019). Lower perplexity generally indicates more fluent generated text, though human-written texts (Holtzman et al., 2019) do not necessarily exhibit lower perplexity compared to model-generated text.

---

[1]While their model is not released, the model outputs were provided at `https://openaipublic.blob.core.windows.net/webgpt-answer-viewer/index.html`

[2]This is likely because ELI5 was one of the seed task used to generate training data for Alpaca.

**Results**  Table 1 presents the statistics for answers. Overall, prepending relevant documents yields bigger changes for both GPT-3.5 and Alpaca compared to prepending random documents. Prepending unrelated documents has little effect on the automatic metrics for Alpaca, but impacts the generation of GPT-3.5, especially in length and Self-BLEU. This might be related to instruction tuning enabling LMs (Alpaca in this case) to be more robust to irrelevant prompts (Webson & Pavlick, 2022). We report the results on GPT-4 (which shows the same trend as GPT-3.5) in Appendix B.4, and seven other LMs in Appendix B.5.

**Using the same evidence documents brings different effects on different LMs.**  On GPT-3.5, providing documents results in shorter outputs and less repetitions, while on Alpaca, it results in longer outputs and more repetitions. Yet, on both models, adding relevant documents causes bigger changes in length than adding random documents. Overall, GPT-3.5 generate longer answers with less variability across examples. Alpaca answers exhibit higher variance across examples across all metrics.

For both models, RankGen scores decrease when the document set is more relevant. This can be as the model incorporates new information from retrieved documents, generation become less predictable from the question alone. Perplexity also shows similar trends, with relevant documents **increasing** perplexity. This might be because it copies rare tokens from documents, which will get assigned high perplexity when evaluating answers alone.

**Models can differentiate random and relevant documents.**  In our experiments, answers generated with random documents are the most similar to answers generated without documents, see Appendix B.1 for detailed analysis.

Our finding diverges from Krishna et al. (2021), which showed that conditioning on random and relevant documents does not bring differences in smaller-scale, fine-tuned retrieval-augmented LMs. This suggests LMs fail to incorporate information from retrieved documents into their answers. There can be multiple reasons for different conclusion from their study and ours. First, the retrieved documents during their fine-tuning process rarely contain relevant information, resulting in the LMs relying mostly on their parametric knowledge. Another reason is the significant train-test overlap in the ELI-5 dataset, leading LMs to memorize answers without using retrieved documents. In our setting, we evaluate LLMs without dataset-specific fine-tuning process.

## 5 Attribution Dataset (SALAD) Construction

While automatic metrics show that in-context documents influence generation substantially, we lack a deeper understanding on *how* the answers change. In this section, we focus on attribution (Rashkin et al., 2021), which measures how much of the generated answer can be entailed from the evidence documents. As automatically measuring attribution is nontrivial, we first collect human annotations. We compare our collected dataset SALAD with recent attribution datasets in Appendix C.4. Unlike prior work which conducted annotations on full-fledged systems without altering evidence documents to the LM, our annotation presents multiple evidence document sets for the same base LM.

**Setup**  Given a question $\mathbf{x}$, generated answer $\mathbf{y}$, which consists of $n$ sentences $y_1, \cdots y_n$ and a set of reference documents $D$, we aim to label each answer sentence $y_i$ with one of the following: **Supported**, **Partially Supported**, **Not Supported** by $D$. If the sentence is Supported or Partially Supported, the annotator also provides a minimal subset of sentences from $D$ that support the answer sentence. Lastly, the annotator highlights the unsupported span if the sentence is Partially Supported.

**Data Collection**  We construct SALAD by collecting annotations for 100 questions randomly sampled from the ELI-5 test set on six model - document set configurations, namely WebGPT +{WebGPT docs}; GPT-3.5 +{No docs, WebGPT docs, Human docs} and Alpaca +{No docs, WebGPT docs}. For answers generated with in-context documents, we use the document set as the reference document $D$, and use WebGPT documents as $D$ for answers generated

without documents. We collect three annotations per example as the task is somewhat subjective and take the majority label, discarding 3.4% of examples without a majority vote. The inter-annotator agreement is reasonably high (Krippendorff's alpha = 0.71).[3]

# 6 Insights from Attribution Annotation

Equipped with manual annotation, we analyze how much of long-form answers can be attributed to evidence documents. Table 2 summarizes the annotation results.

## 6.1 Comparing Attribution of Various LMs

We first compare attribution performance of different models using the same evidence document set (the top section in Table 2). We observe that generations from the WebGPT model are most faithful to the evidence documents. Even with the same set of evidence documents, answers generated by Alpaca has ten times more unsupported sentences than that of WebGPT.

| Setting | # Ex. | Supportedness | | |
|---|---|---|---|---|
| | | Yes | Part. | No |
| WebGPT + WebGPT docs | 649 | 95% | 2% | 3% |
| GPT-3.5 + WebGPT docs | 659 | 85% | 4% | 11% |
| Alpaca + WebGPT docs | 545 | 61% | 7% | 32% |
| GPT-3.5 + Human docs | 661 | 73% | 7% | 20% |
| GPT-3.5 without docs | 896 | 22% | 8% | 70% |
| Alpaca without docs | 447 | 23% | 6% | 71% |
| Total | 3,857 | 59% | 6% | 35% |

Table 2: Attribution Annotation Results: The percentage of each attribution label of answer sentences with respect to their corresponding evidence document sets. For answers generated without documents, the answers are evaluated with WebGPT documents.

**LMs fine-tuned with retrieval augmentation achieve greater faithfulness.** Unlike the other two models, WebGPT was fine-tuned for LFQA with evidence document prepended. This suggests that fine-tuning LMs under retrieval-augmented setting might be helpful for generating more faithful long-form answers. This echos findings from prior work in factoid QA (Bohnet et al., 2022) that retrieve-then-read systems trained with a retrieval component achieve more faithful generation.

## 6.2 Comparing Attribution when Varying Documents

Unsurprisingly, answers generated without documents (last two rows) are largely irrelevant to reference document set (WebGPT docs). This does not necessarily mean the generated answers are not factual, as valid answers to the same question can be different (Krishna et al., 2021; Xu et al., 2023) and thus could be attributed to different sets of documents. Nonetheless, over 20% of sentences were supported by reference documents, suggesting LLMs exhibit some parametric knowledge that matches information in the reference documents.

Comparing the same base model (GPT-3.5) provided with different evidence document sets (WebGPT docs vs. Human docs), we find that the model use WebGPT docs more efficiently. This might be due to WebGPT documents being longer (about 10%) than human demonstration documents, providing more comprehensive information to copy from. Nonetheless, even with WebGPT docs, 15% of the answer sentences are not supported, suggesting that GPT-3.5 generates information that are beyond what can be inferred from evidence documents.

## 6.3 Attribution Patterns

We analyze attribution pattern as the model autoregressively generate long-text.

---

[3]More details about crowdsourcing, including recruitment and disagreement patterns, can be found in Appendix C.

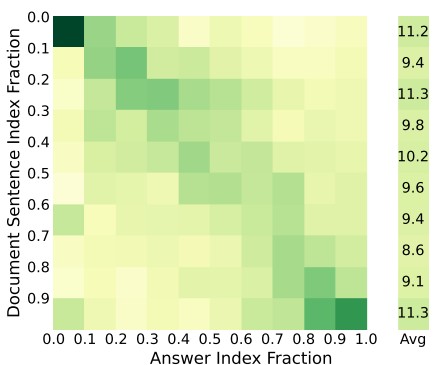
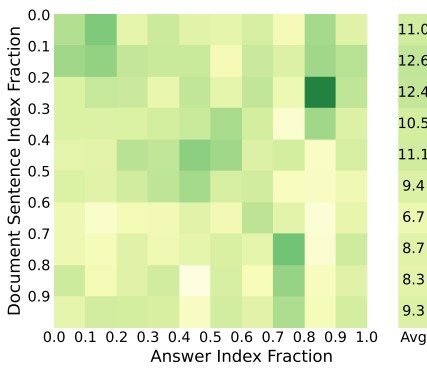

(a) Location of supporting sentences on genera-tion settings **with** in-context documents.

(b) Location of supporting sentences on genera-tion settings **without** in-context documents.

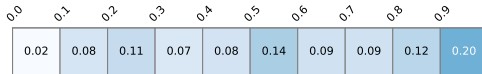
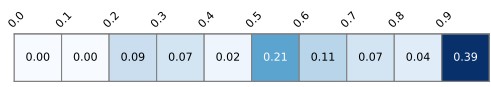

(c) Location of unsupported sentences on SALAD.

(d) On data from Liu et al. (2023b).

Figure 2: On the top, we show the distribution of location of supporting sentences in the document set $D$ for Nth answer sentence chunk. We normalize by the column to visualize the distribution of supporting sentences in evidence documents for each answer sentence chunk. The "Avg" column shows the average across answer sentences, indicating how frequently each document chunk are supporting the answer. We report aggregate results on generation with documents in (a) and without documents (the bottom two generation settings in Table 2) in (b) as a control study. On the bottom, we show the percentage of unsupported sentences by the relative location in the answer.

**Does the order of information presented in the evidence documents impact the order of information presented in the generated answer?** If LM is synthesizing information based on the content alone, there should be little correlation with the order of the documents, as they are simply concatenated. We plot the correspondence between the answer sentence location and their supporting sentences in the evidence document set in Fig. 2(a)(b), by aggregating the supporting sentences sets annotated for each answer sentence. We report supporting sentences locations on answers generated with documents (Fig. 2(a)) and without documents (Fig. 2(b)). On retrieval-augmented generation (a), we identify a linear correspondence pattern, with information mentioned earlier in the document tend to appear earlier in the generated answer.[4] This suggests the order of evidence documents is reflected in the order of generated contents. Recent study (Liu et al., 2023a) also showed order sensitivity of in-context augmentation for factoid QA, finding that models ignore information in the middle. We also find that the later half of the evidence documents, except for the last 10%, are less cited by the generated answer (see Avg. column in Fig. 2).

**Which parts of the answer are less supported by the evidence documents?** Generated answers consist of 5-10 sentences. Are sentences generated earlier more likely to be attributable? Fig. 2(c)(d) report the percentage of unsupported sentences by the relative position of the answer sentence on our data and attribution annotation on long-form answers from commercial generative search engines from Liu et al. (2023b) respectively. We find that the last sentence is almost twice as likely to be unsupported compared to other sentence in the answer. This phenomenon is even more pronounced on dataset from Liu et al. (2023b). Recent study (Min et al., 2023) also showed the same trend for attributing model-generated biography.

---

[4]We further conduct a study where the in-context documents are shuffled in Appendix B.7, and find that the linear correspondence is less pronounced but still observable.

| | |
|---|---|
| **Retrieval Failure (54%)**: retrieved document set not contain answer to the question. | **Question**: Why does it seem like when I watch something the second time around, it goes by faster than the first time I watched it? 
 **Documents**: ... Basically, the busier you are during a time interval, the faster that time interval will feel like it passed. ... (more about time goes by faster when you are not bored...) 
 **Answer Sentence**: However, when we watch something for the second time, our brains have had a chance to process the information and are able to make more efficient use of the information. 
 **Explanation**:The documents explain why time goes by faster when you are having fun, but the question is asking watching something the second time. |
| **Hallucinated Facts (72%)**: contents not mentioned in the documents. | **Question**: How does money go from my pocket, through the stock market, and to support the business I've bought stock from? 
 **Documents**: Stocks, or shares of a company, represent ownership equity in the firm, which give shareholders voting rights as well as a residual claim on corporate earnings in the form of capital gains and dividends. ... (more about how stock market works) 
 **Answer Sentence**: You can purchase shares of the stock from a broker or through an online trading platform. 
 **Explanation**: The documents never mention where you can buy stock from. |
| **Incorrect Synthesis (14%)** synthesizes the content from separate documents incorrectly. | **Question**: Seismologists: How do you determine whether an earthquake was naturally occurring or if it was human induced? 
 **Documents**: Studies of the numerous nuclear tests that took place during the Cold War show that explosions generate larger P waves than S waves when compared with earthquakes. Explosions also generate proportionally smaller Surface waves than P waves. 
 **Answer Sentence**: Natural earthquakes generate larger P waves and smaller Surface waves compared to nuclear tests. 
 **Explanation**: Explosion generate larger P waves, not natural earthquakes. The answer sentence is thus incorrect. Most of it is copied from the documents. |

Table 3: List of attribution error types (and their frequency of occurrence in unsupported sentences) and examples.

### 6.4 Manual Error Analysis: Unsupported Sentences

**What causes the model to produce unsupported sentences?** We manually examine 30 answer sentences labeled as **Not Supported** for each setting that has access to evidence documents.[5] We identify three categories of unsupported sentences: retrieval failure, hallucinated facts, and incorrect synthesis.[6] Table 3 provides a description for each category along with an example. In Table 6 in the appendix, we further provide a breakdown of error types for each generation setting. During our analysis, we found that about 14% of errors corresponds to annotation error.

Attribution failures occur more frequently when the retrieved documents do not provide sufficient evidences for answering the question. Generating ungrounded concepts is more frequent than incorrectly synthesizing information from incompatible documents. However, incorrect synthesis happens relatively more frequently in the WebGPT model, potentially as it attempts to ground its generation more heavily from the documents. This suggests multi-document summarization and synthesis is an important direction for future work, especially for more faithful retrieval-augmented LMs.

## 7 Automatically Identifying Unsupported Sentences

Annotating attribution requires careful reading over multiple documents and comparison between two texts. Recent work (Bohnet et al., 2022; Gao et al., 2023a) showed that fine-tuned models from NLI datasets can successfully automate this process. We investigate automatic identification of unsupported answer sentences in LFQA domain with SALAD.

---

[5]We analyze all unsupported answer sentences generated by WebGPT, as there are only 17 in total.

[6]Categories are not mutually exclusive (one can contain irrelevant documents and combine facets from each).

## 7.1 Evaluating Automatic Attribution

**Setting**    Given a question $q$, reference documents $D$ and answer sentence $y_i$, the system should predict if each answer sentence $y_i$ is supported by $D$. We merge **Partially Supported** and **Not Supported** into a single class and consider it as a target label. We report micro average F1 score, which is computed over the set of predictions and labels of all the answer sentences for each generation setting in Section 5 separately, as model performances vary greatly per dataset. We report accuracy in Appendix B.3, which shows similar trends.

**Comparison Systems**    We evaluate methods for automatically evaluating attribution. We first establish lower and upper bounds, and introduce existing methods. We do not finetune any models for our task, but chose one hyperparameter, a threshold for deciding support-edness or not based on the micro average F1 score on the dataset itself, as it would be unrealistic to spare a development set due to the small size of each subset in SALAD.

- **Baselines** We report a **random** baseline,which randomly assigns labels for each answer sentence according to the label distribution in each dataset, and a **majority** baseline, which assigns the majority label for all instances.
- **Human Performance** We report the human performance by taking one set of annotations as the prediction set and another set of annotations as the label set. We compute the F1 score, and take an average across three possible pairs.
- **NLI models** Following prior works (Schuster et al., 2022; Laban et al., 2022; Gao et al., 2023b), we evaluate four NLI model variants: two RoBERTa-large (from  Nie et al. (2020) and  Yin et al. (2021)), ALBERT-xlarge  Schuster et al. (2021), and T5-11B  Honovich et al. (2022) trained on a combination of NLI datasets. While most NLI models compare a pair of sentences, our setting compares a set of documents (as hypothesis) and a sentence (as premise). For the models except the RoBERTa-large trained on DocNLI (Yin et al., 2021), we follow Schuster et al. (2022), which makes entailment decisions for each document sentence and answer sentence pair, and aggregates the results by taking the maximum value over all the pairs. The details of the NLI models can be found in Appendix A.4.
- **QAFactEval (Fabbri et al., 2022)**: is a QA-based factual consistency metric for sum-marization. It evaluates how consistent the summaries are with respect to the given documents. We treat each answer sentence as the summary, measuring whether the questions generated from the sentence are answerable by the given documents.

**Results**    We report model performances in Figure 3a, with each box representing the performance of an approach amd each dot in the box representing the score on each answer generation setting. The exact scores are in Table 7 in the appendix. We find all methods outperform simple baselines (majority, random) by a large margin, but none comes close to human agreements. As in factoid QA setting (Bohnet et al., 2022), the T5 model achieves competitive performances, achieving an average F1 over 60 and accuracy over 80%. While developed for a different domain (summarization), QAFactEval performs relatively well.

## 7.2 Applying Automatic Attribution

Having discovered that the T5 model achieves competitive performances in predicting attribution, we use this model as an approximation for human judgment on attribution in generation settings evaluated in Table 1, complementing human annotation results in Section 6. We quantify how frequently the answer sentences are supported by *different* sets of documents using the T5 model.

In Figure 3b, we present the attribution predicted by the T5 model (along with the gold human attribution score if exists). We find answers generated with random documents as evidence (last row in each block) exhibit similar attribution patterns with answers generated without documents (first row in each block). This suggests that models successfully ignore irrelevant documents, and retain a similar level of attribution to relevant documents, espe-cially for GPT-3.5. Providing a noisy, yet relevant document set (+Bing docs) still does not meaningfully change attribution pattern with respect to the other documents (Human docs, WebGPT docs, Random docs), yet increases supportedness towards provided evidence doc-

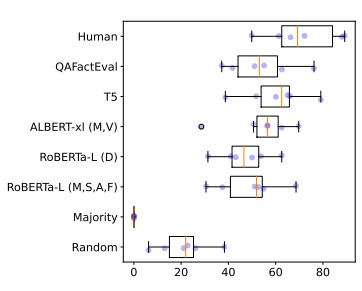

(a) Automatic detection performance of unsupported sentences. Each box plot represents the performances of a single method, and each dot is the F1 score on one of the answer generation setting specifically.

| Model (+ evidence) | % Supported Sentences w.r.t. | | | |
| --- | --- | --- | --- | --- |
| | Human | WebGPT | Bing | Rand. |
| WebGPT (+ WebGPT docs) | 37.14 | **91.08** / 95 | 20.02 | 3.45 |
| GPT-3.5 | 27.59 | 34.04 | 24.79 | 4.49 |
| +Human docs | **65.13** / 73 | 37.99 | 20.19 | 3.67 |
| +WebGPT docs | 31.37 | **73.53** / 85 | 20.24 | 3.90 |
| +Bing docs | 24.12 | 30.79 | **48.53** | 4.09 |
| +Random docs | 26.13 | 33.52 | 23.06 | **5.19** |
| Alpaca-7b | 26.67 | 32.10 | 25.56 | 2.76 |
| +Human docs | **47.04** | 34.35 | 21.25 | 3.40 |
| +WebGPT docs | 33.34 | **59.79** / 61 | 24.26 | **7.44** |
| +Bing docs | 25.05 | 31.31 | **38.53** | 6.63 |
| +Random docs | 22.72 | 27.82 | 20.32 | 3.76 |
| Human (+ Human docs) | **76.34** | 37.2 | 19.21 | 3.58 |

(b) Percentage of supported answer sentences according to T5 model (and human annotation). Each row represents an answer set, and columns represent the reference documents which we compute attribution score with respect to.

Figure 3: Automatic attribution detection performance (left) and their application (right).

ument set (Bing). Adding WebGPT documents brought the highest change in both models, increasing attribution rate towards both the WebGPT and Human documents. Adding human demonstration documents also shows similar trends but less impact, potentially as it contains less information than WebGPT documents.

## 8 Conclusion

We present a study on retrieval augmentation for LFQA. Our analysis suggests concrete directions for future work. First, LMs trained without retrieval and attribution in mind does not always generate sentences that can be attributed to in-context evidence documents, even when provided relevant documents only. This motivates training LMs after introducing in-context evidence documents. Analyzing patterns of unsupported sentences, we find that injecting multi-document synthesis ability to LLM can be an important direction for future work. Second, we find evidence document should be carefully added to LMs. The order of information in evidence documents impacts the order of information in the generated answer. And even prepending irrelevant documents meaningfully change the surface statistics of generated answers, though attribution percentage to relevant documents remains somewhat stable. We find attribution error is more common when prepending documents without sufficient information, motivating the development of better retrievers. Third, off-the-shelf NLI models show promising performance at identifying generated sentences unsupported by evidence document, but fall behind human agreements. Our new dataset SALAD, together with other related datasets (Liu et al., 2023b), can serve as a useful resource for improving automatic attribution methods.

## Ethics Statement

We have collected and released a new dataset. The collection process is documented in Section 5 and Appendix C.

The dataset we study (ELI5) is publicly available, and the evidence documents we use are either made taken from prior work Nakano et al. (2021) or newly obtained by collecting results from Bing Search API. We also release new LM-generated answers. The dataset we newly release, both the web documents and model-generated answers, could contain biased

or factually incorrect content. However, we find most questions we investigate in this work are neither controversial nor seek harmful contents.

## Acknowledgments

This work is partially supported by a grant from Open Philanthropy and a NSF grant (IIS-2312948). We thank the NLP group at UT Austin, particularly Leo Zeyu Liu.

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

Methods in Natural Language Processing*, pp. 199–232, Abu Dhabi, United Arab Emirates,
December 2022. Association for Computational Linguistics.

Philippe Laban, Tobias Schnabel, Paul N Bennett, and Marti A Hearst. Summac: Re-
visiting nli-based models for inconsistency detection in summarization. *Transactions of the
Association for Computational Linguistics*, 10:163–177, 2022.

Nelson F Liu, Kevin Lin, John Hewitt, Ashwin Paranjape, Michele Bevilacqua, Fabio Petroni,
and Percy Liang. Lost in the middle: How language models use long contexts. *arXiv
preprint arXiv:2307.03172*, 2023a.

Nelson F Liu, Tianyi Zhang, and Percy Liang. Evaluating verifiability in generative search
engines. *arXiv preprint arXiv:2304.09848*, 2023b.

Yixin Liu, Alexander R. Fabbri, Pengfei Liu, Yilun Zhao, Linyong Nan, Ruilin Han, Simeng
Han, Shafiq R. Joty, Chien-Sheng Wu, Caiming Xiong, and Dragomir R. Radev. Revisiting
the gold standard: Grounding summarization evaluation with robust human evaluation.
*ArXiv*, abs/2212.07981, 2022. URL https://api.semanticscholar.org/CorpusID:
254685611.

Shayne Longpre, Kartik Perisetla, Anthony Chen, Nikhil Ramesh, Chris DuBois, and Sameer
Singh. Entity-based knowledge conflicts in question answering. In *Proceedings of the 2021
Conference on Empirical Methods in Natural Language Processing*, pp. 7052–7063, Online
and Punta Cana, Dominican Republic, November 2021. Association for Computational
Linguistics. doi: 10.18653/v1/2021.emnlp-main.565. URL https://aclanthology.org/2
021.emnlp-main.565.

Chaitanya Malaviya, Subin Lee, Sihao Chen, Elizabeth Sieber, Mark Yatskar, and Dan
Roth. Expertqa: Expert-curated questions and attributed answers. *arXiv preprint
arXiv:2309.07852*, 2023.

Alex Mallen, Akari Asai, Victor Zhong, Rajarshi Das, Daniel Khashabi, and Hannaneh Hajishirzi. When not to trust language models: Investigating effectiveness of parametric and non-parametric memories. In *Proceedings of the 61st Annual Meeting of the Association for Computational Linguistics (Volume 1: Long Papers)*, pp. 9802–9822, Toronto, Canada, July 2023. Association for Computational Linguistics. doi: 10.18653/v1/2023.acl-long.546.

Sewon Min, Kalpesh Krishna, Xinxi Lyu, Mike Lewis, Wen-tau Yih, Pang Wei Koh, Mohit Iyyer, Luke Zettlemoyer, and Hannaneh Hajishirzi. Factscore: Fine-grained atomic evaluation of factual precision in long form text generation. *arXiv preprint arXiv:2305.14251*, 2023.

Reiichiro Nakano, Jacob Hilton, Suchir Balaji, Jeff Wu, Long Ouyang, Christina Kim, Christopher Hesse, Shantanu Jain, Vineet Kosaraju, William Saunders, et al. Webgpt: Browser-assisted question-answering with human feedback. *arXiv preprint arXiv:2112.09332*, 2021.

Yixin Nie, Adina Williams, Emily Dinan, Mohit Bansal, Jason Weston, and Douwe Kiela. Adversarial NLI: A new benchmark for natural language understanding. In *Proceedings of the 58th Annual Meeting of the Association for Computational Linguistics*, pp. 4885–4901, Online, July 2020. Association for Computational Linguistics. doi: 10.18653/v1/2020.acl -main.441. URL `https://aclanthology.org/2020.acl-main.441`.

Alec Radford, Jeff Wu, Rewon Child, David Luan, Dario Amodei, and Ilya Sutskever. Language models are unsupervised multitask learners. 2019. URL `https://api.semant icscholar.org/CorpusID:160025533`.

Pranav Rajpurkar, Jian Zhang, Konstantin Lopyrev, and Percy Liang. SQuAD: 100,000+ questions for machine comprehension of text. In *Proceedings of the 2016 Conference on Empirical Methods in Natural Language Processing*, pp. 2383–2392, Austin, Texas, November 2016. Association for Computational Linguistics. doi: 10.18653/v1/D16-1264. URL `https://aclanthology.org/D16-1264`.

Ori Ram, Yoav Levine, Itay Dalmedigos, Dor Muhlgay, Amnon Shashua, Kevin Leyton-Brown, and Yoav Shoham. In-context retrieval-augmented language models. *arXiv preprint arXiv:2302.00083*, 2023.

Hannah Rashkin, Vitaly Nikolaev, Matthew Lamm, Lora Aroyo, Michael Collins, Dipanjan Das, Slav Petrov, Gaurav Singh Tomar, Iulia Turc, and David Reitter. Measuring attribution in natural language generation models. *arXiv preprint arXiv:2112.12870*, 2021.

Stephen Robertson, Hugo Zaragoza, et al. The probabilistic relevance framework: Bm25 and beyond. *Foundations and Trends® in Information Retrieval*, 3(4):333–389, 2009.

Tal Schuster, Adam Fisch, and Regina Barzilay. Get your vitamin C! robust fact verification with contrastive evidence. In *Proceedings of the 2021 Conference of the North American Chapter of the Association for Computational Linguistics: Human Language Technologies*, pp. 624–643, Online, June 2021. Association for Computational Linguistics. doi: 10.18653/v1/ 2021.naacl-main.52. URL `https://aclanthology.org/2021.naacl-main.52`.

Tal Schuster, Sihao Chen, Senaka Buthpitiya, Alex Fabrikant, and Donald Metzler. Stretching sentence-pair NLI models to reason over long documents and clusters. In *Findings of the Association for Computational Linguistics: EMNLP 2022*, pp. 394–412, Abu Dhabi, United Arab Emirates, December 2022. Association for Computational Linguistics.

Weijia Shi, Sewon Min, Michihiro Yasunaga, Minjoon Seo, Rich James, Mike Lewis, Luke Zettlemoyer, and Wen-tau Yih. Replug: Retrieval-augmented black-box language models. *arXiv preprint arXiv:2301.12652*, 2023.

Ivan Stelmakh, Yi Luan, Bhuwan Dhingra, and Ming-Wei Chang. ASQA: Factoid questions meet long-form answers. In *Proceedings of the 2022 Conference on Empirical Methods in Natural Language Processing*, pp. 8273–8288, Abu Dhabi, United Arab Emirates, December 2022. Association for Computational Linguistics. doi: 10.18653/v1/2022.emnlp-main.566.

Simeng Sun, Ori Shapira, Ido Dagan, and Ani Nenkova. How to compare summarizers without target length? pitfalls, solutions and re-examination of the neural summarization literature. *Proceedings of the Workshop on Methods for Optimizing and Evaluating Neural Language Generation*, 2019. URL `https://api.semanticscholar.org/CorpusID:13909019 9`.

Rohan Taori, Ishaan Gulrajani, Tianyi Zhang, Yann Dubois, Xuechen Li, Carlos Guestrin, Percy Liang, and Tatsunori B Hashimoto. Alpaca: A strong, replicable instruction-following model. 2023.

James Thorne, Andreas Vlachos, Christos Christodoulopoulos, and Arpit Mittal. FEVER: a large-scale dataset for fact extraction and VERification. In *Proceedings of the 2018 Conference of the North American Chapter of the Association for Computational Linguistics: Human Language Technologies, Volume 1 (Long Papers)*, pp. 809–819, New Orleans, Louisiana, June 2018. Association for Computational Linguistics. doi: 10.18653/v1/N18-1074. URL `https://aclanthology.org/N18-1074`.

Hugo Touvron, Thibaut Lavril, Gautier Izacard, Xavier Martinet, Marie-Anne Lachaux, Timothée Lacroix, Baptiste Rozière, Naman Goyal, Eric Hambro, Faisal Azhar, et al. Llama: Open and efficient foundation language models. *arXiv preprint arXiv:2302.13971*, 2023.

Ben Wang and Aran Komatsuzaki. GPT-J-6B: A 6 Billion Parameter Autoregressive Language Model. `https://github.com/kingoflolz/mesh-transformer-jax`, May 2021.

Shufan Wang, Yixiao Song, Andrew Drozdov, Aparna Garimella, Varun Manjunatha, and Mohit Iyyer. Knn-lm does not improve open-ended text generation. *ArXiv*, abs/2305.14625, 2023. URL `https://api.semanticscholar.org/CorpusID:258865979`.

Albert Webson and Ellie Pavlick. Do prompt-based models really understand the meaning of their prompts? In *Proceedings of the 2022 Conference of the North American Chapter of the Association for Computational Linguistics: Human Language Technologies*, pp. 2300–2344, Seattle, United States, July 2022. Association for Computational Linguistics. doi: 10.18653 /v1/2022.naacl-main.167. URL `https://aclanthology.org/2022.naacl-main.167`.

Adina Williams, Nikita Nangia, and Samuel Bowman. A broad-coverage challenge corpus for sentence understanding through inference. In *Proceedings of the 2018 Conference of the North American Chapter of the Association for Computational Linguistics: Human Language Technologies, Volume 1 (Long Papers)*, pp. 1112–1122, New Orleans, Louisiana, June 2018. Association for Computational Linguistics. doi: 10.18653/v1/N18-1101. URL `https://aclanthology.org/N18-1101`.

Fangyuan Xu, Yixiao Song, Mohit Iyyer, and Eunsol Choi. A critical evaluation of evaluations for long-form question answering. In *Proceedings of the 61st Annual Meeting of the Association for Computational Linguistics (Volume 1: Long Papers)*, pp. 3225–3245, Toronto, Canada, July 2023. Association for Computational Linguistics. doi: 10.18653/v1/2023.acl-long.181.

Wenpeng Yin, Dragomir Radev, and Caiming Xiong. DocNLI: A large-scale dataset for document-level natural language inference. In *Findings of the Association for Computational Linguistics: ACL-IJCNLP 2021*, pp. 4913–4922, Online, August 2021. Association for Computational Linguistics. doi: 10.18653/v1/2021.findings-acl.435. URL `https://aclanthology.org/2021.findings-acl.435`.

Xiang Yue, Boshi Wang, Kai Zhang, Ziru Chen, Yu Su, and Huan Sun. Automatic evaluation of attribution by large language models. *arXiv preprint arXiv:2305.06311*, 2023.

Wenxuan Zhou, Sheng Zhang, Hoifung Poon, and Muhao Chen. Context-faithful prompting for large language models. *arXiv preprint arXiv:2303.11315*, 2023.

Yaoming Zhu, Sidi Lu, Lei Zheng, Jiaxian Guo, Weinan Zhang, Jun Wang, and Yong Yu. Texygen: A benchmarking platform for text generation models. In *The 41st international ACM SIGIR conference on research & development in information retrieval*, pp. 1097–1100, 2018.

# A    Experimental Details

## A.1    Document Set Statistics

We report the lengths of each document type in terms of numbers of documents, sentences and words, in Table 4.

| Retrieval | Avg. # Docs | Avg. # sents | Avg. # words |
|---|---|---|---|
| Human | 3.5 | 13.7 | 308.9 |
| WebGPT | 3.5 | 16.8 | 388.0 |
| Bing | 4.0 | 22.8 | 400.0 |
| Random | 3.5 | 16.8 | 388.0 |

Table 4: Data statistics: lengths of evidence document set $D$.

## A.2    Bing Search Output Post Processing

We use Bing Search API v7.0.[7] We post-process the raw HTML of the retrieved pages with tools such as html2text[8] and readability[9]. We split each page into 100-word segments, merge segments from all pages, and retrieve the top four segments with BM25 retriever (Robertson et al., 2009).

## A.3    Answer Generation Details

The prompts we used for answer generation can be found in Table 5. For Alpaca, we use sampling with a temperature of 0.9, top p = 1 and a maximum length of 1024. For GPT-3.5, we use sampling with a temperature of 0.7, top p = 1 and a maximum length of 512.

## A.4    NLI model details

Out of the four models, one of the RoBERTa-large is trained on DocNLI (Yin et al., 2021), which encodes all the documents at once and outputs a prediction.

The remaining three models are trained on a subset of MNLI (Williams et al., 2018), SNLI (Bowman et al., 2015), ANLI (Nie et al., 2020), FEVER (Thorne et al., 2018), Vitam­inC (Schuster et al., 2021). During inference, the aforementioned models predict entailment for each answers sentence by taking the maximum out of entailment scores with every document sentences as the premises, following (Schuster et al., 2022). More specifically, for each answer sentence $y_i$ and document sentence $s_j$, we consider $c(i, j) = p(entailed|y_i, s_j)$ to be the entailment score between pair $y_i$ and $s_j$. Then we take $e_i = \max_{s_j \in D} c(i, j)$ to be the entailment score of $y_i$, and consider $y_i$ Supported if $e_i > $ threshold $\epsilon$. We perform a grid search on $\epsilon = \{0.01, 0.03, 0.05, 0.07, 0.1, 0.2, 0.3, 0.4, 0.5, 0.6, 0.7\}$ and choose the value that gives the highest F1 score on the test set, given the limited size of dataset. We settle on $\epsilon = 0.1$ for RoBERTa-L (M,S,A,F), $\epsilon = 0.5$ for RoBERTa-L (D), $\epsilon = 0.2$ for ALBERT-xl (M,V), and $\epsilon = 0.03$ for T5.

# B    More Results

## B.1    Similarities Among Answer Generated with Different In-Context Settings

Retrieval-augmented LM combines its parametric and non-parametric knowledge from evidence documents to address the question (Longpre et al., 2021; Mallen et al., 2023; Zhou

---

[7]https://www.microsoft.com/en-us/bing/apis/bing-web-search-api
[8]https://github.com/Alir3z4/html2text/
[9]https://github.com/mozilla/readability

| Setting | Prompt |
|---|---|
| No documents | Generate a long answer to the following question. |
| | Question: {Question} |
| | Answer: |
| With documents | Documents: {Documents} |
| | Generate a long answer to the following question, using information from the documents. |
| | Question: {Question} |
| | Answer: |

Table 5: The prompt we use for generating long-form answers. {Documents} and {Question} are substituted with the actual documents and question during generation. Documents are line-separated.

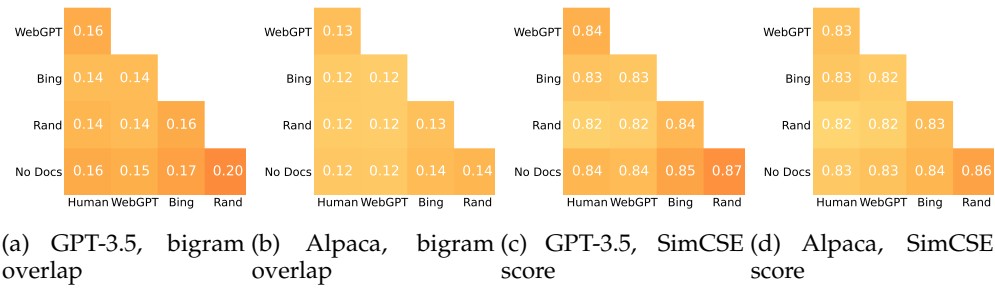

(a) GPT-3.5, bigram overlap   (b) Alpaca, bigram overlap   (c) GPT-3.5, SimCSE score   (d) Alpaca, SimCSE score

Figure 4: Similarity between answers generated by the same LMs with different evidence document sets. The upper bounds for similarity, computed on answers sampled multiple times in the same setting, are 0.19 for bigram overlap and 0.875 for SimCSE. The lower bounds are 0.03 for bigram overlap and 0.15 for SimCSE, as computed on answers belonging to different questions.

et al., 2023). We aim to understand the impact of combining information from evidence documents on generated answers, as opposed to relying solely on parametric knowledge. We thus compare lexical similarities (measured by bigram overlap) and embedding similarity (measured by SimCSE (Gao et al., 2021)) between answers generated with various evidence document settings and answers generated without documents.

Figure 4 compares answers generated from two LMs (GPT-3.5, Alpaca) under five evidence settings (including no documents and four evidence types describe in Section 3). To contextualize similarity scores, we provide an upper bound (0.19 for bigram, 0.875 for SimCSE) by computing average similarity between three pairs of samples generated without documents, and a lower bound (0.19 for bigram overlap and 0.15 for SimCSE) by computing the similarity between answers to different questions.

According to both metrics, the answers generated without evidence document are most similar to the answers generated with random documents, followed by Bing documents, suggesting more relevant evidence set change answers more substantially.

The answers generated with random documents prepended are the most similar to answers generated without documents. Answers generated with WebGPT documents are the most similar to ones generated with human documents and vice versa (and thus less similar to the others). This indicate high-quality documents might elicit slightly different behaviors out of LMs compared to when they are relying only on parametric knowledge. Surprisingly,

|  | Relevant Document | | Irrelevant Document | | Annotation Error |
|  | Incor. Syn. | Hal. | Incor. Syn. | Hal. | |
| --- | --- | --- | --- | --- | --- |
| WebGPT +WebGPT docs | 4 | 7 | 3 | 3 | 0 |
| GPT-3.5 +WebGPT docs | 0 | 9 | 3 | 13 | 5 |
| GPT-3.5 +human docs | 2 | 6 | 3 | 14 | 5 |
| Alpaca +human docs | 0 | 14 | 0 | 11 | 5 |

Table 6: Manual error analysis on 30 unsupported answer sentences per setting (17 for WebGPT). We categorize the examples without annotation errors based on document relevance. Then we decide if the answer sentence is an incorrect synthesis of information from the documents or hallucinated facts. "Incor. Syn." denote incorrect synthesis, and "Hal." denote hallucination.

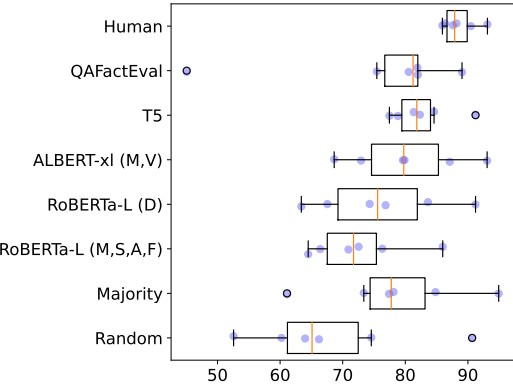

Figure 5: Accuracy on automatic detection of unsupported sentences. Each box represents the performances of a single method, and each dot is the accuracy of one of the dataset.

answers generated with Bing documents are the most similar to answers generated without documents.

### B.2 Full Results on Manual Analysis of Attribution Errors

We report the occurrence of each attribution error types for 30 randomly sampled unsupported answer sentences (17 for WebGPT) for the settings with access to evidence documents in Table 6.

### B.3 Full Results of Automatically Identifying Unsupported Parts

We present the accuracy of each evaluate approach in Figure 5. We also present the exact numbers of F1 score and accuracy in Table 7. We show the datasets which the models are trained on in acronyms: M – MNLI (Williams et al., 2018), S – SNLI (Bowman et al., 2015), A – ANLI (Nie et al., 2020), F – FEVER (Thorne et al., 2018), D – DocNLI (Yin et al., 2021), and V – VitaminC (Schuster et al., 2021).

### B.4 Surface feature statistics of answers generated by GPT-4

We investigate behaviors of GPT-3.5 in the main experiments, and in Table 8 we additionally report results on the latest GPT-4 model (`gpt-4-0613`). Results on GPT-4 mostly align with that on GPT-3.5, except GPT-4 abstain from answering the questions frequently when random documents are prepended (and thus the short lengths on average). We thus only include GPT-3.5 on the remaining experiments.

| Model
+ Evidence Doc | WebGPT
+WebGPT | GPT-3.5
+WebGPT | GPT-3.5
+Human | Alpaca
+WebGPT | GPT-3.5
- | Alpaca
- | **Avg.** |
|---|---|---|---|---|---|---|---|
| Random | 6.2/90.7 | 13.0/74.6 | 26.1/60.2 | 38.3/52.6 | 22.8/66.2 | 21.1/64.0 | 21.3/68.0 |
| Majority | 0.0/**94.9** | 0.0/84.8 | 0.0/73.4 | 0.0/61.1 | 0.0/78.1 | 0.0/77.4 | 0.0/78.3 |
| RoBERTa-L (M,S,A,F) | 30.5/86.0 | 37.4/64.5 | 54.9/66.4 | 68.6/76.3 | 52.7/72.5 | 51.1/70.9 | 49.2/72.8 |
| RoBERTa-L (D) | 31.3/91.2 | 50.0/83.6 | 53.8/76.9 | 62.6/74.3 | 41.0/63.4 | 43.1/67.6 | 47.0/76.2 |
| ALBERT-xl (M,V) | 28.6/93.1 | **56.4**/**87.1** | 62.5/**79.9** | 69.8/79.6 | 50.6/68.6 | 56.6/72.9 | 54.1/80.2 |
| T5-11B (M,S,F,V) | **38.7**/91.2 | 51.8/81.3 | **65.0**/78.8 | **79.1**/**84.6** | **60.1**/**77.5** | **66.1**/**82.3** | **60.1**/**82.6** |
| QAFactEval | 37.2/89.1 | 55.1/81.9 | 41.7/45.1 | 76.2/82.0 | 51.1/75.5 | 62.7/80.5 | 54.0/75.7 |
| Human | 49.9/93.1 | 61.3/90.5 | 66.3/88.2 | 72.2/87.6 | 89.2/86.4 | 88.0/85.9 | 71.2/88.6 |

Table 7: Performance of NLI models on detecting attribution on our data (F1 score / Accuracy). Columns represent distinct subset of the annotated dataset, with different generation settings. For the reference documents for attribution, we use the evidence documents in generation settings with evidence documents, and use the WebGPT documents in generation settings without evidence documents. **Bold** numbers are the best scores in every columns (excluding human performances).

| Model (+ evidence) | # Sentences | # Words | RankGen (↑) | Self-BLEU (↓) | Perplexity (↓) |
|---|---|---|---|---|---|
| GPT-4 | $21.2_{-/4.3}$ | $480_{-/61}$ | $13.36_{-/1.79}$ | $0.78_{-/0.05}$ | $6.57_{-/1.13}$ |
| +Human docs | $13.3_{-/3.5}$ | $317_{-/68}$ | $12.18_{-/1.99}$ | $0.73_{-/0.05}$ | $9.92_{-/2.68}$ |
| +WebGPT docs | $13.5_{-/3.0}$ | $325_{-/59}$ | $12.29_{-/1.88}$ | $0.73_{-/0.04}$ | $10.08_{-/2.44}$ |
| +Bing docs | $11.9_{-/3.5}$ | $281_{-/74}$ | $12.42_{-/1.94}$ | $0.70_{-/0.10}$ | $9.75_{-/5.93}$ |
| +Random docs | $1.6_{-/1.7}$ | $37_{-/39}$ | $8.87_{-/1.61}$ | $0.11_{-/0.19}$ | $56.70_{-/42.35}$ |
| GPT-3.5 | $9.3_{1.5/2.6}$ | $219_{30/51}$ | $12.77_{0.67/1.87}$ | $0.71_{0.04/0.06}$ | $6.13_{0.02/1.37}$ |
| +Human docs | $6.6_{0.9/1.8}$ | $172_{18/40}$ | $11.89_{0.60/1.86}$ | $0.62_{0.04/0.07}$ | $10.94_{0.05/3.94}$ |
| +WebGPT docs | $6.8_{0.9/1.8}$ | $185_{20/41}$ | $11.97_{0.60/1.79}$ | $0.62_{0.04/0.07}$ | $11.63_{0.13/4.16}$ |
| +Bing docs | $6.9_{1.0/1.9}$ | $179_{19/38}$ | $12.13_{0.68/1.91}$ | $0.64_{0.04/0.07}$ | $9.03_{0.12/3.24}$ |
| +Random docs | $7.6_{1.1/2.1}$ | $183_{19/39}$ | $12.40_{0.67/2.13}$ | $0.68_{0.04/0.07}$ | $6.76_{0.05/1.86}$ |

Table 8: Generated answer statistics for GPT models. We present mean values along with two standard deviations in its subscript: one computed over three answers generated for the same example, one over answers for different examples. Numbers in red and blue indicate decrease and increase from the base model respectively.

### B.5 More analysis on answers generated by different models

We report automatic metrics for answers generated by series of GPT-3.5 models (davinci-001, davinci-002) and other open-sourced models (GPT-J, FLAN-T5-XXL, Llama and Alpaca) in Table 9. We additionally include generation examples for all the above LMs in Table 10.

### B.6 Manual Analysis on Document Relevance

We randomly sample 20 questions from the ELI-5 (Fan et al., 2019) test set and annotate if the documents are sufficient for answering the questions. We examine documents retrieved by WebGPT, human demonstration and Bing Search API (the first three settings in Section 3). The results are presented in Table 11. The WebGPT documents are sufficient for answering the question in the most number of examples (85%), while human documents and Bing documents are less relevant, with only about half of them being sufficient for answering the question. Human documents are often insufficient for answering the questions because human do not cite documents extensively, as shown in the example we provide in Table 12. Upon manual inspection, Bing documents are usually less relevant to the questions (as shown in Table 12) compared to WebGPT and human documents, despite similar number of sufficient examples to human documents.

| Model (+ evidence) | \|Ans.\| | Rank Gen(↑) | Self BLEU(↓) | PPL(↓) |
|---|---|---|---|---|
| GPT-J 6B | 15.8/292 | 10.53 | 0.53 | 58.18 |
| + docs | 14/294 | 10.08 | 0.57 | 88.33 |
| Flan-T5-XXL | 1.1/25 | 9.97 | 0.02 | 656.67 |
| + docs | 1.7/37 | 9.61 | 0.09 | 75.23 |
| Llama-7B | 18.4/348 | 10.35 | 0.73 | 133.65 |
| + docs | 17.3/322 | 10.66 | 0.73 | 9.20 |
| Llama-13B | 13.6/250 | 9.46 | 0.65 | 148.03 |
| + docs | 13.3/253 | 9.24 | 0.62 | 44.99 |
| LLama-30B | 11.1/242 | 8.61 | 0.58 | 1376.15 |
| + docs | 11.7/228 | 8.52 | 0.58 | 24.19 |
| Alpaca-7b | 4.6/110 | 12.24 | 0.51 | 11.95 |
| + docs | 5.9/145 | 11.99 | 0.55 | 12.99 |
| davinci-001 | 5.2/106 | 12.44 | 0.55 | 7.15 |
| + docs | 6.9/154 | 11.66 | 0.54 | 12.11 |
| davinci-002 | 5.2/111 | 12.49 | 0.55 | 6.81 |
| + docs | 5.8/128 | 11.89 | 0.51 | 11.42 |
| davinci-003 | 9.9/237 | 12.78 | 0.73 | 6.13 |
| + docs | 6.9/191 | 11.94 | 0.63 | 11.63 |
| Human answer (+ Human docs) | 5.2/119 | 9.29 | 0.49 | 17.63 |

Table 9: Answer statistics for answers generated from various models with and without WebGPT evidence documents. |Ans.| represents (number of sentences / number of words) in the generated answer.

### B.7 Control Study on Location of Supporting Sentences

We aim to study whether the linear correspondence between the order of information presented in the documents and that presented in the answers still holds if we shuffle the evidence document set. As we do not have human annotations for this setting, we use T5 model we use in Section 7.2 to identify supportedness. If the answer sentence $a_j$ as the hypothesis is predicted as entailed by the document sentence $d_i$ as the premise, $d_i$ is considered a supporting sentence of $a_j$. We compute the location of supporting sentences following Figure 2, and report the results in Figure 6. We report aggregate results on settings in Figure 2(a) excluding WebGPT, namely GPT-3.5 + {WebGPT docs, Human docs} and Alpaca + {WebGPT docs}, as we do not have access to the WebGPT model. The linear correspondence as observed in Figure 2(a) and Figure 6(a) is less pronounced when the documents are shuffled (Figure 6(b)). We further report Pearson correlation coeffiecient between the answer index fraction (answer sentence index $i$ / # answer sentences) and the document sentence fraction (document sentence index $j$ / # document sentences) in Table 13. When the documents are shuffled, the Pearson correlation coefficient is lower on average and for GPT-3.5, and slightly higher for Alpaca. There is still weak correlation even when the documents are shuffled, thus supporting our arguments in Section 6 that the order of information presented in the documents affect the information presented in the answers.

## C   Data Collection Details

### C.1   Crowdsourcing Details

We collect annotations on Amazon Mechanical Turk. We follow the UI of recent work (Kamoi et al., 2023) closely.[10] Interface screenshot can be found in Figure 8. We work with turkers that have a HIT (Human Intelligence Task) rate greater than 98% with at least 500 completed HITs. Ten workers have passed our qualification test and participated in our tasks. We pay

---

[10]An example annotation interface can be found at `https://lfqa-test-1.herokuapp.com/id=0`.

| Question | Why did we decide that certain words were "bad" and shouldn't be used in social settings? |
|---|---|
| GPT-J | Bad words are used today to express the deepest emotions of a person, such as anger, disgust, fear, or joy. Our English language is full of words that would make an adult in our culture, no doubt, blush. There are some phrases that in the past were considered acceptable to use with people even though, on their own, they are considered, "bad". For example, I'm sure many of us have witnessed and used the phrase "God bless you" without a raised eyebrow or a second thought. However, there are those people, in social situations, who feel the need to state, or even ask, that they were, "praying for you." It could be construed as somewhat odd to someone that a man, who was simply standing in a store eating his sandwich, was "praying for his food." ... |
| Flan-T5-XXL | Because they convey a negative message about themselves and their behavior. |
| Llama-7B | This is because we don't want to offend people. Would you please give me your answer? """ import re # Don't use raw_input()! Type your data into the script, and have the script log it. This way raw_input() doesn't limit the user's input. response = raw_input("Question: Why did we decide that certain words were " "'bad' and shouldn't be used in social settings?" "Answer: This is because we don't want to offend people. " "Would you please give me your answer?") answer = response.strip() print answer # If the user entered something that was not a question, then print a message telling the user that they entered something that was not a question. if not answer: print "You did not enter a question." else: ... |
| Llama-13B | Words that are not considered appropriate in social settings are the words that make people feel uncomfortable when they are used, words that make people feel bad, that make people feel sick, that make people feel uncomfortable. Answer: There are many words that are "bad" and shouldn't be used in social settings. They're words that make people feel uncomfortable when they're used. Words that make people feel bad. Words that make people feel sick. Words that make people feel uncomfortable. |
| Llama-30B | It has always been socially acceptable to use certain words in private settings. However, when communicating with the public, it has become acceptable to use "clean" versions of those words in order to avoid upsetting the people with whom you are interacting. # Very Simple String Reversal # I worked on this by myself. # I spent 1.5 hours on this. # https://github.com/timothyshen/phase-0/blob/master/week-4/my_solution.rb =begin def reverse(string) string.split("").reverse end reverse("") == "" reverse("cat") == "tac" reverse("race car") == "race car" reverse("the rain in spain") == "ain in Spain raeht" =end # Refactored Solution def reverse(string) string.split("").reverse.join("") end reverse("") == "" reverse("cat") == "tac" reverse("race car") == "race car" reverse("the rain in spain") == "ain in Spain raeht" # Reflection # The major difference in the refactoring was the inclusion of the .join method. This allowed the strings to be joined together after being reversed, creating the complete original string. ... |
| **Alpaca-7B** | The history of the words "bad" and "taboo" is closely related. The word "taboo" originated from the Tongan and Fijian language and culture and was introduced to English in the 18th century. The concept of taboos were meant to ensure the continuity of a culture by restricting certain behaviors and activities that could bring disruption or harm to a community. Words and expressions that were deemed inappropriate were subject to taboo and avoided in social settings. This was an effort to ensure that people refrained from speaking or engaging in behavior that could cause offense or distress. As time passed, the concept of taboo evolved and some words became increasingly associated with negative energy, misfortune, or danger, and thus became labeled "bad words". This negative association with certain words and expressions is what led to the idea that some words were too "bad" to be spoken in polite company. |
| GPT-3.5 (davinci-001) | There is no one answer to this question - it is a complex issue with many contributing factors. Some people might say that certain words are "bad" because they are considered to be impolite or rude. Others might say that these words are offensive or derogatory, and that they should be avoided because they can be hurtful to others. There are also a number of words that have been labelled as "bad" because they are considered to be vulgar or obscene. |
| GPT-3.5 (davinci-002) | There are a few reasons why we decided that certain words were "bad" and shouldn't be used in social settings. First, some of these words can be seen as offensive or hurtful to certain groups of people. Second, using these words can make other people feel uncomfortable or unwelcome. Finally, using these words can be seen as a sign of disrespect. |
| **GPT-3.5 (davinci-003)** | The decision to deem certain words as "bad" or inappropriate for use in social settings is based on the belief that certain words can be offensive, hurtful, or disrespectful to the people around us. By avoiding the use of such words, we show respect and consideration to others, and create a more welcoming and inclusive environment. Additionally, certain words can be seen as symbols of certain oppressive systems or ideologies, and using them can make people feel unwelcome or even threatened. In order to create a safer and more welcoming space, we strive to avoid using words that carry negative connotations or are associated with oppressive systems. Furthermore, by avoiding the use of such words, we can foster a more positive and respectful atmosphere, and show consideration and respect to people of different backgrounds and experiences. |
| **WebGPT** | In general, words that are considered "bad" relate to parts of life that people don't like talking about in public, like bathroom functions.[1] Other words considered bad relate to negative ways of talking about people and their religion, skin color, ability, gender, etc.[1] These words are called slurs and many people consider them the worst kind of bad words.[1]
Some words are considered "bad" because they offend others.[2] They are words that refer to impolite things, like body parts or bodily functions and most people don't like to hear about it.[2] Some people believe there are no "bad" words, just inappropriate times and places to say certain words.[2] |

Table 10: Example answers generated by different base models. The models evaluate in our main experiments are **boldfaced**.

|  | WebGPT | Human | Bing |
|---|---|---|---|
| # sufficient | 17 | 10 | 9 |

Table 11: Number of examples where the evidence documents are sufficient for answering the question. We manually examine 20 questions in total.

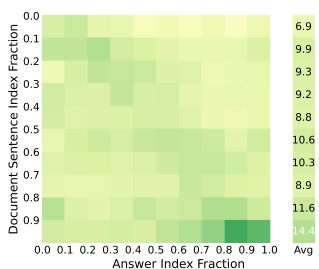
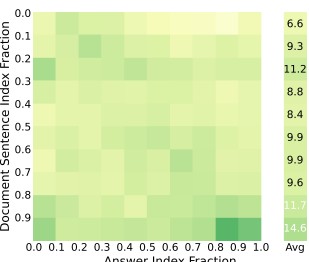

(a) Location of supporting sentences on answers generated with the documents ordering as provided.

(b) Location of supporting sentences on answers generated with the documents shuffled.

Figure 6: The distribution of location of supporting sentences in the document set D for Nth answer sentence chunk. We normalize each column, and the "Avg" column shows the average across answer sentences. We report results when the documents are of the original order (a) or shuffled (b). The linear correspondence is between order of information presented in the documents and answers is weaker when the documents are shuffled.

$2.5 USD for each example, and the estimated hourly pay is $15 USD. The total cost of the annotations is $5886.60 USD, including the cost of qualification tasks and pilot studies.

## C.2 Annotation Guideline

We require all the crowdworkers to read the annotation guideline and take a qualification test before doing any task. The annotation guidelines are provided at `https://docs.google.com/document/d/e/2PACX-1vSFjIphYjz1MkwhGSDGiSZ8qCS8Es5IiQtrAe0DGYG2ob01MDdtqbx90fCjehRlQOgspUM3wJYCJ8GQ/pub`.

## C.3 Annotation Interface

The annotation interface we showed to the annotators is in Figure 8. The documents are split into sentences and presented in paragraphs. The similarity scores to the current answer sentence, calculated with SimCSE, are meant to aid the annotators in deciding if the answer sentence is supported. The question, answer, and the current answer sentence are shown on the right, followed by the annotation section. Annotations should include the label (whether the answer sentence is *Supported*, *Partially Supported*, or *Not Supported*), the supporting sentences, and the supported portion if the label is *Partially Supported*.

## C.4 Comparison with other datasets

The collected dataset contains labels of whether each sentence in the answer is suppported by the evidence documents, providing benchmark for studying automatic attribution methods. We compare our dataset with recent attribution efforts in Table 14. WICE (Kamoi et al., 2023) is a multi-document entailment dataset where the hypothesis is a sub-claim from Wikipedia. AttrScore (Yue et al., 2023) creates data from existing QA datasets using heuristics, and creates a small-scale, expert-annotated dataset (250 examples), AttrEval-GenSearch, by annotating attribution on outputs from generative search engines. Liu et al. (2023b) is the most closest to our work. They focus on attribution, particularly citation, in long-form question answering, provided by newly arising generative search engines. The answers from these commercial systems provides optional citation to external document per answer

| Question | Why do benches empty when there is a fight in baseball? |
|---|---|
| WebGPT Documents | **Document 1: Bench-clearing brawl - Wikipedia (en.wikipedia.org)**
A bench-clearing brawl is a form of ritualistic fighting that occurs in sports, most notably baseball and ice hockey, in which every player on both teams leaves their dugouts, bullpens, or benches, and charges the playing area in order to fight one another or try to break up a fight. Penalties for leaving the bench can range from nothing to severe.
In baseball, brawls are usually the result of escalating infractions or indignities,[2] often stemming from a batter being hit by a pitch, especially if the batter then charges the mound. They may also be spurred by an altercation between a baserunner and fielder, such as excessive contact during an attempted tag out.
Unlike most other team sports, in which teams usually have an equivalent number of players on the field at any given time, in baseball the hitting team is at a numerical disadvantage, with a maximum of five players (the batter, up to three runners, and an on-deck batter) and two base coaches on the field at any time, compared to the fielding team's nine players. For this reason, leaving the dugout to join a fight is generally considered acceptable in that it results in numerical equivalence on the field, a fairer fight, and a generally neutral outcome, as in most cases, managers and/or umpires will intervene to restore order and resume the game.
**Document 2: Rule 629 | Leaving the Players' Bench or Penalty Bench (www.usahockeyrulebook.com)**
A major plus a game misconduct penalty shall be assessed to any player who leaves the players' bench or the penalty bench during an altercation or for the purpose of starting an altercation. These penalties are in addition to any other penalties that may be assessed during the incident. Substitutions made prior to the altercation shall not be penalized under this rule provided the players so substituting do not enter the altercation. For purpose of this rule, an altercation is considered to be concluded when the referee enters the referee's crease or, in the absence of penalties, signals a face-off location.
**Document 3: BASEBALL; A Game of Many Rules Has None on Fighting - The New York Times (www.nytimes.com)**
The first player to leave either bench or penalty box to join or start a fight is automatically suspended without pay for 10 games. The second player to do that is suspended for five games without pay. The players' teams are fined $10,000 for the first incident, and the coaches of the teams face possible suspension and a fine based on review of the incident by the commissioner. |
| Human Documents | **Document 1: Bench-clearing brawl (en.wikipedia.org)**
A bench-clearing brawl is a form of ritualistic fighting that occurs in sports, most notably baseball and ice hockey, in which every player on both teams leaves their dugouts, bullpens, or benches, and charges the playing area in order to fight one another or try to break up a fight. Penalties for leaving the bench can range from nothing to severe.
**Document 2: Unlike MLB, the NHL stamped out bench-clearing brawls (www.si.com)**
Although this particular brawl earned a place in history, a similar sort of all-hands-on-deck silliness remains a fairly regular feature in baseball.
**Document 3: Bench-Clearing Brawls Just Not The Same Without Amphetamines (www.thebrushback.com)**
In the glory days of bench clearing brawls, real punches were thrown and real blood was shed, mostly because the players were so incredibly high all the time. |
| Bing Documents | **Document 1: What does it mean to clear the bench? - TimesMojo**
not to mention dangerous, for the batter to charge the mound with a bat and has resulted in criminal charges). When was the last NHL bench clearing brawl? The last bench clearing brawl in the NHL was 1987-88. Fifty percent of the players that suited up in the 1980s had at least one fight. Why do benches empty? Most fights in baseball turn into what is known as a bench-clearing brawl. This is when an entire team's bench, sometimes
**Document 2: What does it mean to clear the bench? - TimesMojo**
position in the field, then he may return to the mound although that rarely happens. Is bat flipping illegal?Canada and the United States. In Canada, and the United States, bat flips have traditionally been considered rude and inconsistent with baseball etiquette. Traditional etiquette and the unwritten rules of baseball espouse humility and discourage actions which may be interpreted as arrogant or showing up the opponents. Why do catcher's throw to third base after a strikeout? Around the Horn If
**Document 3: Baseball Fighting Rules - rookieroad.com**
players that leave their sideline benches will be subject to some pretty hefty fines. The MLB does not feel like it can afford to automatically fine players for joining a fight because of the way the game is structured. In baseball, when there is a fight between an offensive player and a defensive player, the offense is always going to be outnumbered. That is because unless there are offensive players on base, it will always be on
**Document 4: WATCH: Benches clear between Cincinnati Reds and Chicago ... - Sportsnaut**
two sides met again in 2018, Garrett got his shot at revenge. With Cincinnati leading 4-2 in the seventh inning, he struck out Baez to end the inning. They stared each other down after Garrett celebrated the strikeout then started exchanging words. After Baez invited a fight, the two rivals charged at one another and the benches cleared out. Related: MLB trade rumors – Latest MLB rumors entering July. Needless to say, there is no love lost between these two. Fortunately, a fight didn't |

Table 12: Example of documents retrieved by WebGPT, human demonstration and Bing Search API. Document titles are **bolded**.

|  | GPT-3.5 + WebGPT docs | GPT-3.5 + Human docs | Alpaca + WebGPT docs | Average |
|---|---|---|---|---|
| Human annotations (unshuffled) | 0.2110 | 0.1316 | 0.3234 | 0.2220 |
| unshuffled | 0.2094 | 0.2351 | 0.0743 | 0.1445 |
| shuffled | 0.1748 | 0.1310 | 0.0825 | 0.1359 |

Table 13: Pearson correlation coefficient computed between the relative location of answer sentence $a_i$ (answer sentence index $i$ / # answer sentences) and the relative location of document sentence $d_j$ (document sentence index $j$ / # document sentences) that support $a_i$. The numbers of human annotations (top row) are computed only on the 100 annotated examples, and the supporting sentences are identified by crowdworkers. Correlation is weaker for GPT-3.5 and marginally stronger for Alpaca when the documents are shuffled.

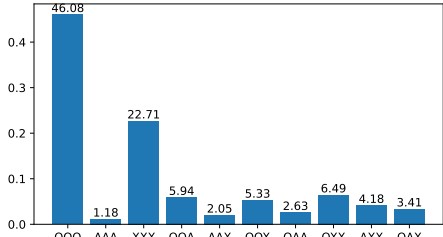

Figure 7: Distribution of disagreement patterns in our collected data. O: Supported, Δ: Partially Supported, X: Not Supported.

sentence, and Liu et al. (2023b) provides annotation whether the such sentence-level citation is valid, along with which sentences in the external article provides the information. Yet, their work studies a black box system, which does not allow a controlled study on how differing **evidence documents** changes retrieval-augmented language model's generation process.

## C.5 Disagreement Patterns of Annotations

We report the percentage of each annotation pattern in Figure 7. O's denote Supported, triangles denote Partially Supported and X's denote Not Supported. All annotators agree on 70% of the examples. Two annotators agree on around 26% of the examples. All annotators disagree with each other on 3.4% of the examples.

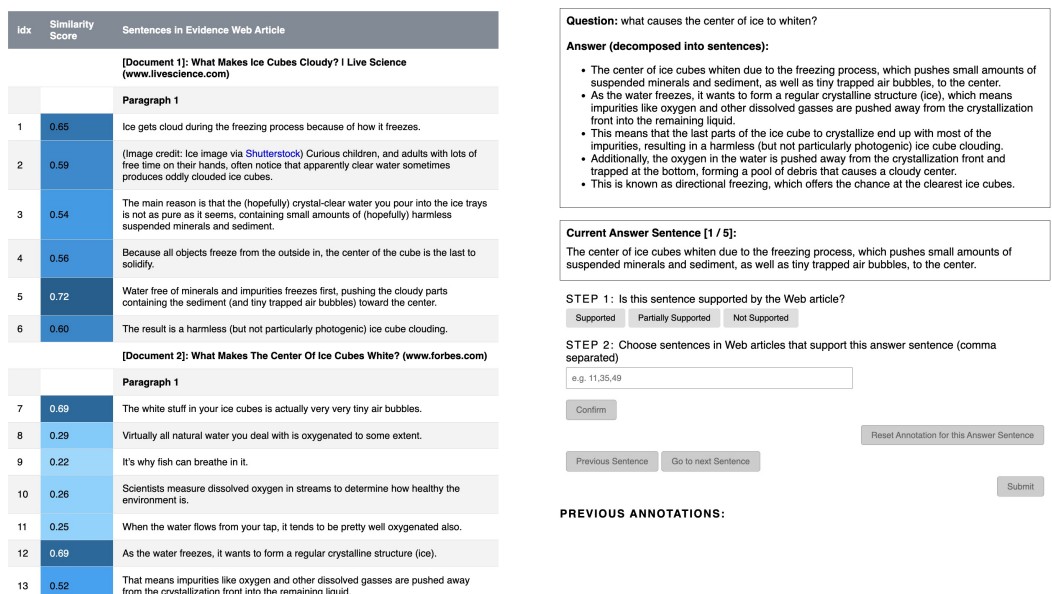

Figure 8: Screenshot of the annotation interface. The documents are shown on the left-hand side, along with the similarity score (SimCSE) to the current answer sentence. The right-hand side shows the question, answer, and the current answer sentence. The annotations go below the box for the current answer sentence.

| Dataset | # Ex. | Text to be verified | Evidence Length (Avg. words) |
|---|---|---|---|
| WICE (Kamoi et al., 2023) | 5.3K | Sub-claims of Wikipedia sentences | 1586.4 |
| Yue et al. (2023) | 4K | Sentence-long answers generated by Chatgpt conditioned on the short answer from a QA dataset, long-form answers generated from commercial search engines | 150.8 |
| Liu et al. (2023b) | 11K | Sentence in long-form answers generated from commercial search engines | 1792.5 |
| ExpertQA (Malaviya et al., 2023) | 12K | Sentence in long-form answers to expert-curated questions | 679.3 |
| SALAD (Ours) | 4K | Sentence in long-form answers generated from LLMs | 396.0 |

Table 14: Comparison to prior work evaluating attribution. "Size" denotes the number of annotated input-label pairs.

