# OpenReview forum: "Understanding Retrieval Augmentation for Long-Form Question Answering"
_colmweb.org/COLM/2024/Conference — COLM_

### Official Review · Reviewer_pLGb · 2024-05-08

**Rating:** 7
**Confidence:** 3
**Ethics Flag:** 1

**Summary:**

This paper looks at how in-context examples affect long-form answers (more than a word or phrase) in LLM generation.  The authors vary both the model and the documents given, and measure aspects of the answer.  They find that the answers are impacted by the documents given (even by random documents), and answers are often related to the input documents (even corresponding to the order of the documents given), but answers are not always attributable to the documents (unless the model was trained for that behavior).  They evaluate automatically using several statistics that indicate higher quality (e.g., looking at fluency, length, variance), but no automatic metric is used that really targets a correct answer.

This paper covers and interesting topic and delves into it a bit, but the description of the task and the dataset (SALAD) leaves me a bit unclear.  Are the answers in the SALAD dataset given scores of correctness or just how much their answer is supported by the source document?  I would like to understand exactly what is in the SALAD dataset and how it is useful.  Is it useful for evaluating new LLMs or only the LLMs used to generate it?

Notes:
- Figure 1 references sections 3 and 5 for attribution, but I think it should be sections 5 and 7 (as stated in the caption to the same Figure)
- Figure 2 - the heat maps are hard to interpret.  What are they intended to show... just an idea of approximately where the answers are coming from?  Also the text of parts (a) and (b) runs together... would be more readable if spaced farther apart.

[After author discussion]
I am raising my score to 7 - Good paper, accept

**Questions To Authors:**

- Are the answers in the SALAD dataset given scores of correctness or just how much their answer is supported by the source document?  I would like to understand exactly what is in the SALAD dataset and how it is useful.  Is it useful for evaluating new LLMs or only the LLMs used to generate it?

- What are Figure 2's heat maps intended to show?

**Reasons To Accept:**

- Empirical research in LLM generation with in-context documents

**Reasons To Reject:**

- Lack of clarity with methodology and dataset [Addressed in Author Discussion]

- Weak automatic evaluation method for most LLMs (only WebGPT, GPT-3.5, and Alpaca get human evaluations)

---

> ### Author Rebuttal · Authors · 2024-05-31
>
> Thank you for your review and encouraging feedback. We address your questions and concerns below.
>
> Re: What is evaluated in the SALAD dataset?
> We clarify here that the goal of our paper is to measure attribution, instead of answer correctness, which we have attempted to clarify in the paper:  In the “Setup” paragraph, we mention that labels of a particular answer sentence is “Supported”, “Partially Supported”, and “Not Supported”. The labels are referring to whether this answer sentence is “supported” by the reference documents, and we mention nothing about correctness of the answer.
>
> Re: Is SALAD useful for evaluating new LLMs?
> The SALAD dataset only evaluates attribution of the LLMs selected, and cannot provide attribution evaluation on newly generated answers. Yet, SALAD could serve as a benchmark for “evaluating” automatic attribution. This is what we do in Section 7. Once we identify a good automatic attribution prediction model (e.g. the T5 model which we reported performance in Section 7.1), we could use that model to evaluate new generations. We will integrate this discussion into the main text in the revised version.
>
> Re: Figure 1 reference
> Yes, it should be Section 5 and 7, thank you for catching this! We will update the figure in the camera-ready version.
>
> Re: Figure 2
> They are intended to show which part of the answers are supported by which part of the documents. As mentioned in the paper, the heatmap has higher numbers on the diagonal, showing that the order of information mentioned in the answer is roughly in line with the order of information mentioned in the documents. In other words, answer sentences that come first are supported by sentences that appear earlier in the documents, and sentences that come later are supported by sentences that appear later in the documents. We hope this is clear, and we could clarify if there are further questions.
>
> Re: Automatic evaluation instead of human evaluation for a subset of LMs
> As mentioned in the rebuttal to review x7LT, the cost for human annotation for this project (100 examples per setting for 6 settings) is almost $6k USD. This is the most we can have under our budget, and we believe that the automatic evaluation method reasonably correlates with human annotation and that the results/discussion in Section 7.2 would not change much with more human evaluation.

---

> > ### Comment · Reviewer_pLGb · 2024-06-05
> > **Thank you for clarification**
> >
> > I am clearer on the paper now, I will raise the score to 7.

---

### Official Review · Reviewer_vo4B · 2024-05-09

**Rating:** 6
**Confidence:** 4
**Ethics Flag:** 1

**Summary:**

This paper presents a comprehensive analysis and study on retrieval-augmented long-form question answering (LFQA). Different from short answers, the LFQA receives less study on how the retrieved documents affect the long-form answers in various perspectives, which is the focus of this work. Through extensive experiments, the authors made several observations including different LMs' capabilities towards attributing their generated answers to the retrieved contexts, the effect of different sets of documents under retrieval for common metrics such as perplexity, self-BLEU, etc, and comparing different automatic models for fine-grained attribution evaluations.

Overall, this paper is well written and clearly delivers the objective. The LFQA problem is indeed an under-explored area with much room for improvement. The findings given from this work provide useful insights and suggest potential directions for future work. The authors also conduct a comprehensive study with extensive experiments, testing various perspectives ranging from different LMs, different sets of retrieval corpus, diverse metrics, and attribution evaluations, which make the conclusion more convincing.

A few limitations are observed:
1. Some of the experiments/analysis are not so constructive and hence are less meaningful from my understanding. For example, Table 1 compares RankGen which focuses on the correlation between the question only and the generation. This does not seem to be relevant under the retrieval-augmented setting, because the answer could incorporate more contents from the context, making it less corresponding with the question. And the results do show this pattern, without any surprise. Further, two of the metrics in Table 1 demonstrate inverse effect given by relevant documents compared to no or random documents. I do not see how this could bring any meaningful insights from these results. Other more relevant metrics could be considered such as measuring the coherence, coverage, diversity etc.
2. The last observation in Section 4 could be made more precise and meaningful - The divergence between this work and Krishna et al. (2021) is probably due to the model capacity. One possible conclusion made is that higher-capacity LMs are more robust to random contexts and pay more attention to relevant context. To better show this point, it could be more interesting to evaluate smaller LMs for emprirical comparison.
3. Some experiments in Table 2 also seem less meaningful. I don't see the value of conducting experiments without documents and evaluate the supportedness. First, the documents from WebGPT might not be relevant for the question. Second, since the documents are not presented, it is not fair to force the model to generate answers relevant to the documents.
4. The setting in section 7.1 merge partially supported with not supported. This is not entirely intuitive to me. Is there any reason why the partially supported should be treated as the negative class for NLI? What if this is merged with the supported label?

**Questions To Authors:**

The setting in section 7.1 merge partially supported with not supported. This is not entirely intuitive to me. Is there any reason why the partially supported should be treated as the negative class for NLI? What if this is merged with the supported label?

**Reasons To Accept:**

The problem of LFQA this work studies is an interesting and under-explored area.
This work provides a comprehensive evaluation and analysis, revealing many interesting observations which could be meaningful for further studies.
The authors also introduce a valuable dataset annotated with fine-grained attribution labels in the sentence level. This could be useful for future research.

**Reasons To Reject:**

Some of the experiments and conclusions are not constructive.
Some experimental settings and observations are not clearly explained.
The dataset being evaluated is less diverse.

---

> ### Author Rebuttal · Authors · 2024-05-31
>
> Re: Table1 metrics
> Measuring coherence, coverage, diversity can be more insightful. Self-BLEU is designed to measure diversity. Coherence and coverage are difficult to evaluate automatically. We also argue that RankGen measures relevance of the generated answer with respect to the question, which is important for studying retrieval augmentation. Consistent RankGen scores across various document settings show that models are able to ignore irrelevant documents, while staying relevant to the question. We believe that these results still bring some insights for the state-of-the-art models.
>
>
> Re: Bigger vs. smaller models
> We agree that it might be interesting to show that larger models are more robust to random contexts and better incorporate relevant contexts! Our focus is on evaluating LLMs that are relevant (at the point of writing) and cover various properties (open/closed-source, trained with/without retrieval component). We will add discussion on the size of the model as future work.
>
> Re: Table 2 Results
> We present experiments without documents as a control setting. Comparing this setting with all the other settings with documents, one could conclude that the models are incorporating information from the documents into the answers. We will make arguments clearer.
>
> Re: Merging “Partially Supported” with “Not Supported”
> Both “Partially Supported” and “Not Supported” sentences contain information that is not supported by the documents. If the automatic attribution model detects any information that is not mentioned in the documents, the model would predict the sentence as NOT “Supported”.
>
> Re: Dataset being evaluated is less diverse
> We only evaluate model generated answers on the ELI-5 dataset, which is the principal long-form QA dataset. While it would be nice to include questions from domains other than community QA, yet it will require collecting such questions, as well as high quality documents, which is expensive and out of scope for our paper.  Concurrent work (Malaviya et al., 2023) collected expert-curated questions across various domains and evaluated attribution of answers generated by LMs. Their settings differ from ours as they studied different attribution strategies (e.g. directly generating citation, post-hoc retrieval). We focus on a specific strategy, i.e. “retrieve-and-read” and present careful controlled study with different types of retrieved documents.
>
> Reference:
> Malaviya et al, 2023 https://arxiv.org/abs/2309.07852

---

> > ### Comment · Reviewer_vo4B · 2024-06-05
> > **Response to authors’ rebuttal**
> >
> > I appreciate the authors’ replies to all my questions. I will keep my original evaluation and am leaning positive towards the acceptance of this paper.

---

### Official Review · Reviewer_4Hj1 · 2024-05-10

**Rating:** 7
**Confidence:** 4
**Ethics Flag:** 1

**Summary:**

This works presents a study on retrieval augmentation for long-form question answering. They perform 2 controlled studies : first they fix the LM and varying evidence documents, second they fix evidence documents and varying the LMs. This is a good setup as it resulted in interesting results/discussion useful for a better understanding of how retrieval augmentation work.

They also present a new dataset -SALAD, containing human annotations of sentence-level answer attribution in long-form question answering.

**Questions To Authors:**

1. Why do you think some of your results differ significantly from other works? Is it because of a different setting or some limitations of earlier works that you solved?

**Reasons To Accept:**

1. The experiments are very extensive and well-detailed.
2. Some of the findings are very informative and novel such as answer length decreases and answers become concise when relevant documents are given as evidence. The findings that "order of evidence documents is reflected in the order of generated contents" and "the last sentence is almost twice as likely to be unsupported compared to other sentence in the answer" are very interesting and provides a new depth to the understanding of RAGs
3. The conclusion suggests strong directions for future models using RAGs and can be very useful for the community
4. Overall I think this paper definitely improves the understanding of RAGs and provides new insights into their limitations and how dependent they are to given documents

**Reasons To Reject:**

1. They authors argue that their results differ significantly from earlier works such as Krishna et al. I would have liked a better argument as to why that is happening and we should trust their results as compared to previous works. It might leave readers in a dilemma of whose results to trust.
2. The discussion of results such as in Section 6 is difficult to follow. I understand that the reviewers had a lot of points to discuss and space constraint might have reduced some discussion, but a better written discussion section would help the readers get the key takeaways from this work more easily
3. Figure 2 plot captions are difficult to read and a better readability will def help

---

> ### Author Rebuttal · Authors · 2024-05-31
>
> Thank you for your review and encouraging feedback. We address your questions and concerns below.
>
> Re: Results differ from findings from Krishna et al.
> This is a valid concern! We do think results from both studies are trustworthy, but the difference in settings and models leads to different conclusions.
>
> Krishna et al proposes two possible reasons for why LMs are not using retrieval documents:
> The retrieved documents have little overlap with the gold answer, so the generation model ignores the documents.
> There is significant train-test overlap in the ELI-5 dataset, and models could answer based on their parametric knowledge alone, without the help of retrieved documents.
> Both reasons do not apply in our setting (documents are more relevant and models are not fine-tuned on the data from the same distribution). There are other differences:
> First of all, they are only measuring ROUGE-L, which is the overlap between the predicted answers and the gold answers. However, we measure the attribution of the answers, which is how much the answers are supported by the prepended documents.
> The model architectures are significantly different. They use a weaker retriever (REALM, Guu et al., 2020) compared to WebGPT (whose retriever is trained specifically for this task, †leveraging a commercial search engine, Bing), leading to documents potentially of less quality. They also use a much weaker LM (486M parameters) to generate the answer, compared to GPT-3.5 (175B parameters). Both the better quality of retrieved documents and a much stronger generation model could have contributed to the differing results.
>
> We will update the paper to further discuss the differences between our set-up and theirs.
>
> Re: Better written discussion section
> We thank the reviewer for the feedback. While we  have included paragraph titles for a few of our analyses to improve clarity, we will clearly highlight takeaways with paragraph titles in the revised version for better readability.
>
>
> Re: Better figure 2 plot captions
> We apologize for the harder-to-follow caption. The caption is intentionally shortened due to the length limitation, and we will update it in the camera-ready version.
>
> Reference:
> Guu, K., Lee, K., Tung, Z., Pasupat, P. &amp; Chang, M.. (2020). Retrieval Augmented Language Model Pre-Training. Proceedings of the 37th International Conference on Machine Learning Available from https://proceedings.mlr.press/v119/guu20a.html.

---

> > ### Comment · Reviewer_4Hj1 · 2024-06-05
> >
> > I thank the reviewers for their response and the paper would definitely merit from including discussion regarding Krishna et al. I am inclined towards keeping my score and think this paper should be accepted.

---

### Official Review · Reviewer_x7LT · 2024-05-12

**Rating:** 7
**Confidence:** 4
**Ethics Flag:** 1

**Summary:**

The paper focuses on the topic of retrieval-augmented long-form question answering (LFQA). The two controlled studies show how the evidence documents or the LLMs impact the generated answers. The performance is evaluated in terms of both surface features and attribution to the evidence documents. It also includes an human-annotated dataset named SALAD, which could facilitate the research in LFQA. Several key findings help the committee better understand the status of retrieval augmentation for LFQA.

**Questions To Authors:**

1. Will you open-source the dataset?
2. How do you know GPT3.5  trained without retrieval and attribution? Is it a guess?

**Reasons To Accept:**

1. The paper presents a controlled study with varying evidence documents and LLMs for the retrieved-augmented LFQA task.The study allows a clearer understanding of the impact of each component.

2. It introduces the SALAD dataset with human attribution annotations for LFQA.

3. The analysis and evaluation are easy to understand and thorough. It summarizes different error types as well as methods for automatic identification of unsupported answers.

**Reasons To Reject:**

1. The size of the dataset is relatively small, which might affect the conclusion of this paper.  How do you select the 100 questions? The question types could affect the results as well.

2. According to the results in FIgure 3b, it is interesting  that the results have a large variance when comparing Human docs and WebGPT docs. It seems Human only works well with Human docs, so does WebGPT (WebGPT docs) and Bing (Bing docs).

---

> ### Author Rebuttal · Authors · 2024-05-31
>
> Thank you for your review and encouraging feedback. We address your questions and concerns below.
>
> Re: Sample selection / Data set size
> We select the 100 questions randomly from the test set of ELI-5, where we have WebGPT predictions available (a total of 271 questions). ELI-5 dataset contains diverse topics and so does the sampled set upon manual inspection. Our labeled data set is relatively small because of the cost of human annotations – the total cost is almost $6k USD for this number of questions. This covers 100 answers and around 700 sentences for each system, resulting in around 4k sentences that could be used for evaluation of automatic attribution.
>
> Re: variance when comparing different reference documents
> This is definitely an interesting result, but this should not be considered as a weakness of the paper. In figure 3b, the attribution numbers are computed when the documents specified with the ‘+’ sign are prepended. Figure 3b indicates that the percentage of supported documents is high only when the attribution is computed with respect to the prepended documents.
>
> Re: Open-source
> Yes, we will open-source the dataset upon publication.
>
> Re: Is GPT-3.5 trained without retrieval and attribution?
> Thank you for pointing this out! We do not know for sure it is trained without a retrieval component. However, none of their technical reports or documentation (e.g. Ouyang et al., 2022) has mentioned a retrieval component. We will update the text to reflect that this is not a known fact.
>
> Reference:
> Ouyang, L., Wu, J., Jiang, X., Almeida, D., Wainwright, C., Mishkin, P., ... & Lowe, R. (2022). Training language models to follow instructions with human feedback. Advances in neural information processing systems, 35, 27730-27744.
>
> GPT-4 technical report. https://arxiv.org/pdf/2303.08774

---

> > ### Comment · Reviewer_x7LT · 2024-06-06
> > **Response**
> >
> > Thanks for the clarification.

---

### Author Response · Authors · 2024-06-05
**Gentle Reminder for Discussion**

Hi! We thank all the reviewers for your reviews and hope that you have read our rebuttal. Do you have any thoughts regarding our rebuttal?

---

### Decision · Program_Chairs · 2024-07-10

**Decision:**

Accept

**Comment:**

The paper explores how retrieval-augmented language models (RAGs) utilize evidence documents for long-form question answering (LFQA). The study introduces the SALAD dataset, containing human annotations for sentence-level answer attribution, and evaluates existing methods for automatic attribution judgment. The findings reveal that while LMs can leverage relevant documents, the generated answers are only partially attributable to the documents, with unsupported sentences being a significant issue.

The paper is well-executed and offers valuable insights into the use of RAGs for LFQA. Despite some concerns about the clarity of the discussion and the relevance of certain experiments, the comprehensive study and novel findings contribute significantly to the field. Therefore, the paper is accepted with minor revisions.

To enhance the paper, the authors should provide a clearer explanation of why their results differ from earlier works like Krishna et al., ensuring readers understand and trust their findings. The discussion section should be rewritten for better clarity, making the key takeaways more accessible. Improving the readability of Figure 2 captions is also recommended.